# TEM Investigation of Asymmetric Deposition-Driven Crystalline-to-Amorphous Transition in Silicon Nanowires

**DOI:** 10.3390/ma15207077

**Published:** 2022-10-12

**Authors:** Yuan Zang, Lianbi Li, Jichao Hu, Lei Li, Zelong Li, Zebin Li, Song Feng, Guoqing Zhang, Caijuan Xia, Hongbin Pu

**Affiliations:** 1Department of Electronic Engineering, Xi’an University of Technology, Xi’an 710048, China; 2School of Science, Xi’an Polytechnic University, Xi’an 710048, China

**Keywords:** silicon nanowires, electron microscopy, phase transformation, bending, semiconductors

## Abstract

Controlling the shape and internal strain of nanowires (NWs) is critical for their safe and reliable use and for the exploration of novel functionalities of nanodevices. In this work, transmission electron microscopy was employed to examine bent Si NWs prepared by asymmetric electron-beam evaporation. The asymmetric deposition of Cr caused the formation of nanosized amorphous-Si domains; the non-crystallinity of the Si NWs was controlled by the bending radius. No other intermediate crystalline phase was present during the crystalline-to-amorphous transition, indicating a direct phase transition from the original crystalline phase to the amorphous phase. Moreover, amorphous microstructures caused by compressive stress, such as amorphous Cr domains and boxes, were also observed in the asymmetric Cr layer used to induce bending, and the local non-crystallinity of Cr was lower than that of Si under the same bending radius.

## 1. Introduction

As one of the most important types of semiconductor nanowires (NWs), Si NWs have been widely employed in nanotechnology applications such as nanosized transistors [1], actuators [2], battery anodes [3], and power sources [4]. The mechanical properties and structural transformations of Si NWs under external stress are critical for their safe and reliable use and for exploring novel functionalities of various types of nanodevices. Significantly, exposed to bending strain, NWs exhibit novel properties and functions, such as decreased conductance, reduced bandgap, and broadened near-edge emission in the cathodoluminescence spectra [5,6,7,8,9].

Si NWs have been shown to possess room-temperature nanoscale plasticity [10], which may result from the formation of dislocations with possible unconventional slip systems [11]. At high stresses, a crystalline-to-amorphous (c–a) transition was observed under uniaxial tension, compression, and bending [11,12,13,14,15,16]. This amorphization appears to be substantially influenced by the radius and orientation of the NW, as well as by experimental conditions such as strain rate, shear stress, and temperature [13]. Previous studies have reported the transmission electron microscopy (TEM) characterization of NWs bent by mechanical compression and shear forces [14,15,16,17,18], which is of great significance for the study of dislocation-controlled phase transitions and/or plasticity in bent Si NWs. However, additional components are employed to generate mechanical stress in bending experiments, such as colloidal films [14], hydrogen silsesquioxanes [17,18], and nanoprobes [15,16], which makes the bending process relatively complex; as a result, bent Si NWs are difficult to be directly applied in electronic devices and other systems owing to the interfere of additional components.

In recent years, asymmetric deposition [19,20,21,22] has emerged as a popular technique for the controlled bending of NWs. This approach is usually based on pulsed laser deposition [20,21] and molecular beam epitaxy [19,22], which have been employed to create bent NWs of GaAs and ZnO. We previously demonstrated a new asymmetric deposition process to bend Si NWs into a desired shape using sideways electron-beam evaporation (EBE) and manufacture arc-shaped composite nanowire arrays [23]. Based on our observations, asymmetric deposition of metallic materials such as Cr and Ti can bend Si NWs in the direction of the e-beam deposition, and the bending radius can be adjusted by varying the deposition angle of the sample stage and the deposition thickness. Compared with the bending of NWs using assisted apparatuses, this bending process is more compatible with the manufacturing of microelectronic devices, EBE is a common process in Si foundry. Furthermore, the metallic materials controlling the bending form an integrated system with the NWs, and then construct the device as an important component, which is different from the external force-driven bending of NWs with assisted apparatuses [14,15,16,17,18]. However, the atomic-scale characterization of bent Si NWs during asymmetric deposition has not been reported to date. Herein, the asymmetric deposition-driven amorphization during bending of Si NWs is characterized using atomic-resolution TEM. We also discuss the relation between forward bending and microstructure of the Si NWs, which is of great significance to further explore the mechanism of NW bending driven by asymmetric deposition. In addition, we thoroughly investigate the phase transition of the metallic materials utilized to trigger the bending process, owing to its critical influence on the physical and electrical properties of asymmetrically deposited heterojunctions.

## 2. Experimental Section

Low-pressure chemical vapor deposition (LPCVD) was used to create Si NWs on an n-type on-axis Si (111) substrate. SiCl_4_ was employed as a precursor for the growth of Si NWs. The carrier gas was H_2_ (15%) in argon, which was flown through the Si precursor bath and into the growth chamber. The diameter and position of the Si NWs were defined using Au colloids, which were immobilized on the substrate using a previously described technique [24]. Si NWs fabricated using the above method produced monocrystalline wires with <111> preferred orientations, as shown in Appendix A. Because the feature size in our experiments is 100 nm~200 nm in diameter, the Si NWs are in an unconfined nanoscale state and thus prone to dislocation-governed deformation. In fact, dislocations did control the initial deformation of our Si NWs. An electron-beam evaporator (RME-E2000) was used for asymmetric growth, as shown in Appendix A. It is believed that the forward bending of the NWs is related to the thermal and/or lattice mismatch between Si and the metallic materials. The lattice constant of Cr is much smaller than that of Si, while the coefficient of thermal expansion is 2.5 times that of Si. On the contrary, the lattice constant of Sn is large, while the coefficient of thermal expansion is small, as shown in Appendix A. It can be inferred from experiments that lattice mismatch or thermal mismatch is the main reason for bending. Therefore, as the folding trigger, the metallic materials such as Cr and Sn were deposited on the side wall of vertically oriented Si NWs by using the same technique. Si NWs with different bent strain *ε*_Bent_ are prepared by varying the deposition thickness and NWs’ diameter (Figure 1). TEM (Talos L120C) was employed to characterize the asymmetric deposition-driven crystalline-to-amorphous transition in Si NWs. The microstructure and compositional distribution of the Si NWs were also investigated by Scanning TEM (STEM) coupled with energy dispersive X-ray spectroscopy (EDX).

## 3. Results and Discussion

Figure 2a,b shows bright-field images of a bent Si NW with an asymmetric Cr layer. The maximum bent strain in the Si NW can be calculated as 2.7%, according to the standard equation *ε*_Bent_ = *r*/(*r* + *R*)%, where *R* and *r* are the radius of the bending curvature and Si NW, respectively. Owing to the thickness/mass dependence of electronic scattering, the high-angle annular dark field (HAADF) STEM image in Figure 2c reveals increased contrast between the two regions. The lower region appears brighter because of its rich Cr content. The asymmetrically deposited Cr layer on the Si NW also presents different contrast between the upper and lower regions, owing to the different thicknesses caused by the asymmetric deposition, as shown by the STEM-EDX elemental mapping images in Figure 2d,e. The Si NW exhibited a [111] orientation with a *d*-spacing of 3.14 Å. The tensile strain on the upper Si surface resulted in the Si (111) crystal lattice being rotated with an angle of ~5°, as confirmed by the two adjacent Si (111) diffraction spots (Figure 2f, inset). According to the lattice calculations, a 5° orientation rotation can release 0.38% of the tensile strain in the Si (111) crystalline planes, as demonstrated in Appendix A. Distortion occurred in some regions of the crystalline-Si (c-Si) lattice, forming amorphous-Si (a-Si) nanodomains (Figure 2g,h). No other intermediate crystalline phases were observed during the c–a transition, indicating the occurrence of a direct phase transition from the original c-Si lattice to the amorphous phase. Misfit dislocation (MD) nucleation and interaction in Si NWs were also involved in ultra-large bent straining, as shown in Figure 2i,j. We assume that structural defects such as MDs released the initial strain, with amorphization occurring only at higher local strains.

The Cr layer exhibited an uneven surface with periodic fluctuations of 10–30 nm, owing to the release of surface compressive strain, as shown in Figure 2k. The fast Fourier transform (FFT) patterns (Figure 2k, inset) and the HRTEM image in Figure 2l clearly indicate that the Cr layer had a body-centered cubic (bcc) structure with [110] orientation and a *d*-spacing of 2.03 Å. High-density stacking faults (SFs, Figure 2k) and MDs (Figure 2m) were also observed. Further straining produced “amorphous boxes” along the [1,2,3,4,5,6,7,8,9,10] direction, resulting in numerous c-Cr/a-Cr interfaces, as shown in Figure 2n. These amorphous boxes became larger as the expanding amorphous pockets collided with one another (Figure 2o). The HRTEM images of the Si/Cr interface are shown in Figure 2p,q. At the Si/Cr interface, the *d*-spacings of both Si(111) and Cr(110) diverged from their typical values and approached one another. This was mainly caused by interfacial mismatch-induced stress, which was greatly reduced due to the formation of the MDs [23]. Note that the lattice misfit of the Si(111)/Cr(110) heterostructure is about 35.0%, as calculated from the *d*-spacings and FFT patterns, which is given by *ε* = (*ɑ*_Si_ − *ɑ*_Cr_)/*ɑ*_Si_. The lattice mismatch at the Si(111)/Cr(110) heterostructure is greatly reduced to 1.6% in this region with the MDs. Although the lattice mismatch has been greatly reduced, the forward bending of Si NWs is still caused by the mismatch stress, as demonstrated in Appendix A.

Figure 3a shows a bent Si NW subjected to 10.3% bending strain. The HAADF (Figure 3b) and EDX elemental mapping (Figure 3c,d) images of the bent Si NW show that the Cr layer grew asymmetrically on the Si NW. As shown in Figure 3f, a Cr layer with a thickness of 2.8 nm was also deposited on the upper surface of the Si NW, which was uneven due to tensile strain. The Si NW exhibited a c–a transition, as also confirmed by the FFT patterns in the inset of Figure 3f. It is worth noting that the Si NW with 10.3% bending strain has higher non-crystallinity, and only a few nanosized Si crystalline grains remain. In addition to the TO mode of Si nanocrystals at 516.6 cm^−1^, an obvious broad band around 480 cm^−1^ characteristic for a-Si (Appendix A) is also observed in Raman spectra, confirming the c–a transition [25]. The amorphous degree observed by HRTEM is higher than that measured by Raman spectroscopy (>22%), which may be due to the larger scattering volume of the Raman spectrum than that characterized by HRTEM, and the bent NWs are only partially amorphous. Furthermore, the 514 nm laser in Raman spectroscopy still causes crystallization [26], although the characterization process is relatively short. The diffraction contrast of a-Si generated via asymmetric deposition-driven amorphization differs from that of a-Si vitrified from a Si melt, revealing that their structures are distinct. The c-Cr/a-Si interface is shown in Figure 3g. Cr(110) columnar grains with a *d*-spacing of 2.03 Å are clearly observed in Figure 3h,i. High-density MDs appeared in the Cr asymmetric layer (Figure 3j). Compared with the Si NW with a larger bending radius, the amorphization of the Cr layer was more significant. A large number of amorphous microstructures, such as a-Cr domains (Figure 3k) and a-Cr boxes (Figure 3l), were also observed; these are typical features of stress-induced amorphization. Crystal streaks with obvious structural features corresponding to c-Cr were still present, separated in space by “a-Cr boxes”. Furthermore, under the same bending radius, the Si NWs were almost completely amorphous, whereas the local non-crystallinity of Cr was less than 30%, indicating that the c-Cr had a high strain capacity and could relieve more strain through MDs and SFs. This is also because Cr with a smaller feature size could withstand relatively low bending strain values under the same bending radius.

To further investigate the growth mechanism of asymmetric deposition-induced bending, Si/Sn heterostructures were also characterized by HRTEM, as shown in Figure 4a,b. The Sn layer was composed of high-density crystalline grains with a [211] orientation and a *d*-spacing of 2.02 Å (Figure 4c,d). In contrast with the asymmetric Cr deposition on Si, the asymmetric deposition of Sn did not contribute to the bending of Si NWs, which is consistent with the SEM results shown in Appendix A. This is due to the fact that lattice strain can be released by discrete Sn grains (Figure 4a), and the local crystal orientation of these grains is not completely consistent, resulting in an inhomogeneous strain distribution and making oriented bending difficult. The bending process is related to physical properties such as the lattice constant, orientation, and thermal expansion coefficient of the interface. In addition, the growth mode of the asymmetric deposition layer also plays an important role in it.

## 4. Conclusions

In conclusion, Cr was found to be capable of bending Si NWs in the direction of the e-beam deposition. The thermal/lattice mismatch between Si and Cr caused forward bending of the Si NWs. The symmetric deposition-driven amorphization of Si NWs was directly observed with atomic resolution. The solid-state amorphization involved a direct transition from the crystalline to the amorphous phase through the accumulation of defects. No other intermediate crystalline phase was observed prior to amorphization. Some regions of the c-Si lattice were distorted under a bending strain of 2.7%, forming a-Si nanodomains, while a Si NW with a higher bending strain of 10.3% had a higher degree of non-crystallinity, with more than 98% of the c-Si becoming amorphous. We believe that the present technique can be generally employed to control the shape and internal strain of NWs, with potential applications in the fabrication of novel nanodevices.

## Figures and Tables

**Figure 1 materials-15-07077-f001:**
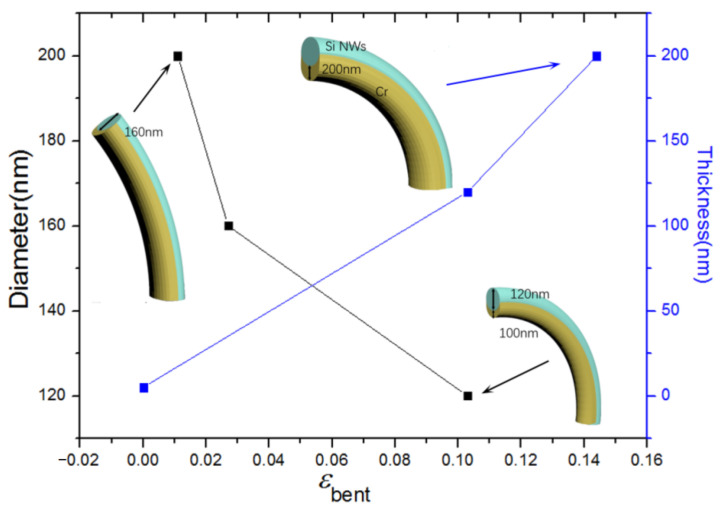
Si NWs prepared by asymmetric deposition with different bent strain *ε*_Bent_.

**Figure 2 materials-15-07077-f002:**
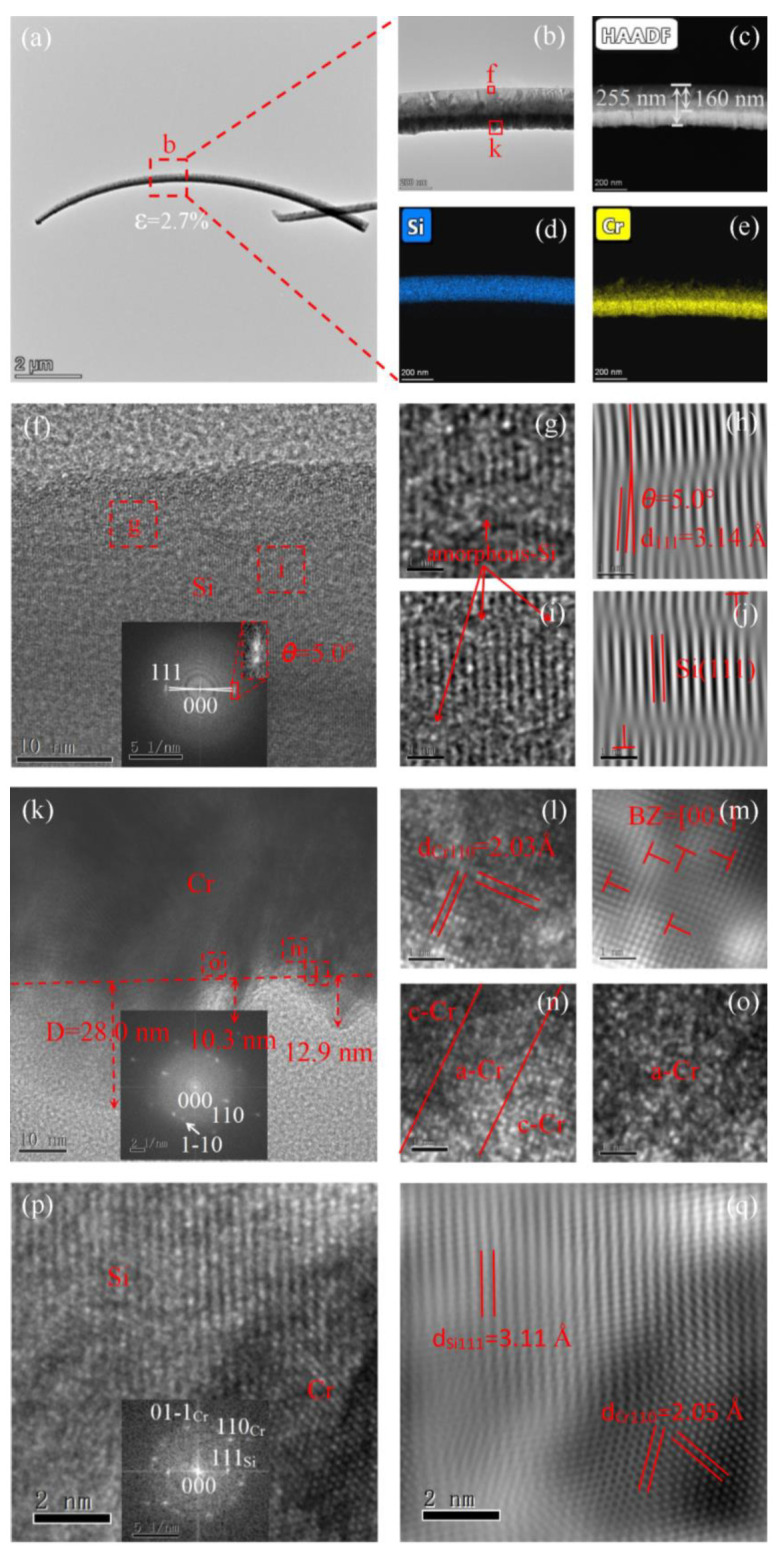
Bright-field TEM (**a**,**b**), HAADF (**c**), and STEM-EDX elemental mapping (**d**,**e**) images of Si NWs with 2.7% bending strain. HRTEM image of Si (**f**–**j**), asymmetric Cr (**k**–**o**), and Si/Cr interface (**p**,**q**). Panels h, j, m, and q show Fourier-filtered images. The insets of panels f, k, and p display FFT patterns of Si, asymmetric Cr, and Si/Cr interface, respectively.

**Figure 3 materials-15-07077-f003:**
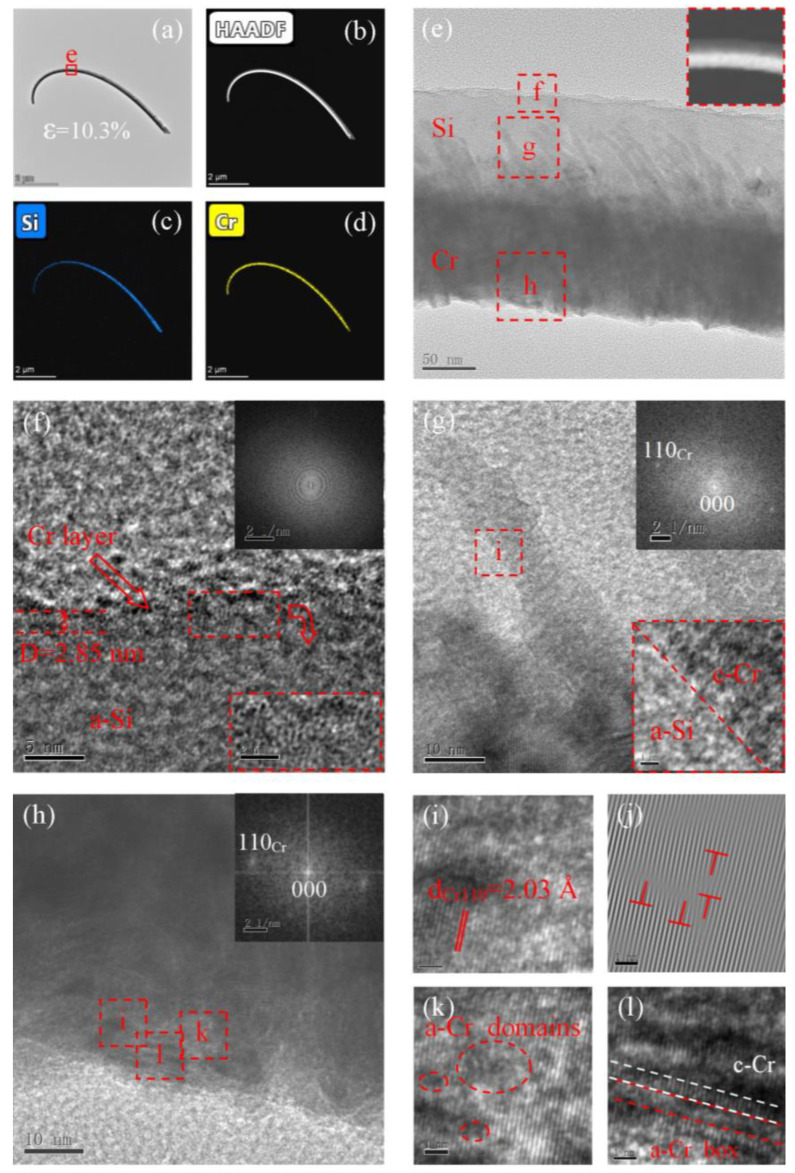
Bright-field TEM (**a**,**e**), HAADF (**b**), and STEM-EDX elemental mapping (**c**,**d**) images of Si NWs with 10.3% bending strain. HRTEM image of Si (**f**), Si/Cr interface (**g**), and asymmetric Cr (**h**–**l**). Panel (**j**) shows Fourier-filtered images. The insets of panels f–h display FFT patterns of Si, Si/Cr interface, and asymmetric Cr, respectively.

**Figure 4 materials-15-07077-f004:**
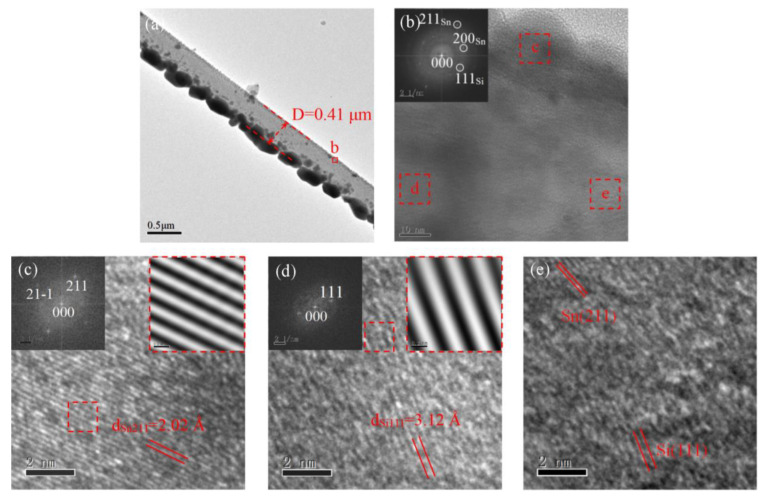
Bright-field TEM (**a**,**b**) of Si NW with asymmetric Sn layer. HRTEM image of asymmetric Sn (**c**), Si (**d**), and Si/Sn interface (**e**). The insets of panels b–d display the FFT patterns and Fourier-filtered images.

## Data Availability

Not applicable.

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
