# Peer review of "TEM Investigation of Asymmetric Deposition-Driven Crystalline-to-Amorphous Transition in Silicon Nanowires"

_materials, 2022, doi:10.3390/ma15207077_

Round 1
Reviewer 1 Report
- Figure 1 should be on one page instead of spread between 2
-
INcrease the font of Si and Cr in the elementa mapping images 1d 1e and 2c 2d
-
Mention how the elemental mapping was done i.e figure description should say STEM-EDX
-
In illustration (cartoon) of difference in fig1, fig 2 and fig 3 deposition in the results and discussion could help make the paper clearer i.e how the interlayer is different in the three scenarios leading to different microstructures
-
Introduction can include a line or two on how microstructure influences the growth mechanism
-
Experimental procedure should include the type and model of TEM used.
- Can include elemental mapping of Sn/Si in fig 3 if available
-
A plot of bending strain calculated in the paper with the three different condition can help visualise the effect of interlayer.

Author Response
We sincerely appreciate reviewer’ kind comments and suggestions on our manuscript. Please see the attachment. We have carefully revised the manuscript.
We hope that the changes made will give satisfaction and the revised version meets the standards of Materials.
Sincerely yours,
Lianbi Li

Reviewer 2 Report
The authors present a microscopic study using electronic microscopy with different magnification and another complementary technique such as EDS to reveal the effect of Cr and Sn deposition on the bending of silicon nanowire. This work is interesting and can be publishable after completing these comments.
1-Please mention more details concerning the novel properties and function in the introduction ‘’NWs exhibit novel 29 properties and functions when exposed to bending strain (Hsin et al., 2008; San Paulo et 30 al., 2007; Han et al., 2009; Qin et al., 2008)’’
2-could you please explain why the bent si NWs fabricated reported elsewhere could not be applied directly in electronic devices, see lines 46,47.
3-please provide more details concerning the advantageous of your technique regarding the others, especially, when you present a comparison, see lines 59,60.
4- Why did you use Cr and sn for the asymmetric growth?
5- you mentioned a rotation of Si (111) crystal lattice of 5 ° and you mentioned 50° in figure 1 i), please correct the value.
6-How did you estimate a tensile strain of 0.38% released in the si (111) crystalline planes?
7- is there a way to provide some statistical study that shows the percentage of amorphous region regarding the crystalline region?
8-In the line 126, you mentioned (The HRTEM images of the Si/Cr interface are shown in Fig. 1(p,q).) Where is the Fig. 1 (p.q), maybe I’m wrong.
9-in lines 166-168, you mentioned that “This is due to the fact that lattice strain 166 can be released by discrete Sn grains, and the local crystal orientation of these grains is 167 not completely consistent” Could you please show on the figure where these grains are located?
10-The resolution of the figures in the manuscript is good, but, you need write the label correctly, because readers can be confused between the label and the region names.
11- Please add more references
12-I see a third department in the SI, is it true?

Author Response

(The authors gave the same response as above.)

Reviewer 3 Report
Referee report
The article “TEM investigation of the asymmetric-deposition-driven crystalline-to-amorphous transition in silicon nanowires” is devoted to attractive and actual topic – growth of Si nanowires. crystallization of amorphous Si films using laser annealing. Despite the fact that researchers have studied these phenomena many years, it is still actual problem. The authors also touched upon an interesting problem - how mechanical deformations affect the crystallization and amorphization of silicon. It is known that the kinetics of first order phase transitions depends on the difference in the specific Gibbs potentials (specific enthalpy) of a substance in the amorphous and crystalline states. It is known that mechanical deformations contribute to the elastic energy, and hence to the specific enthalpy, see for example the work (M.D.Efremov, et. al. Excimer laser and RTA stimulation of solid phase nucleation and crystallization in amorphous silicon films on glass substrates. J.Phys.:Condens. Matter 8, 1996, pp.273-286) and references in it. I think the article can be interesting for researchers dealing with amorphous/polycrystalline Si nanowires. It can be published after minor revision.
I have some comments and remarks:
1) In many figures, there is no letter corresponding to them, for example, in figures 1a, 1f, 1k, as well as in many figures from series 2. Figure 3a is absent at all, although it is mentioned in the text.
2) It is not clear from text how the authors had determined that “more than 98% of the c-Si becoming amorphous”. For example, according to Raman spectroscopy data, it is possible to do the quantitative analysis of the phase ratio of nanocrystalline/amorphous silicon, see for example the work – Zhigunov et. al. On Raman Scattering Cross Section Ratio of Crystalline and Microcrystalline to Amorphous Silicon. Appl. Phys. Lett., v.113, pp. 023101(1-4), (2018). The authors should explain how they determined the quantitative phase composition from electron microscopy data. As an advice for the future, we can recommend the authors to use the fast and non-destructive method of Raman spectroscopy.
Recommendations: accept with minor revision
Author Response

(The authors gave the same response as above.)

Reviewer 4 Report
The manuscript" TEM investigation of asymmetric deposition-driven crystal line-to-amorphous transition in silicon nanowires" is well presented and analyzed. Can be accepted in its present form.
Author Response
We sincerely appreciate reviewer’ kind comments on our manuscript. We hope that the changes made will give satisfaction and the revised version meets the standards of Materials.
Sincerely yours,
Lianbi Li
Round 2
Reviewer 2 Report
The authors have replied to my comments. Nice work and congratulations!